# Statistical and Machine Learning Modeling of Psychological, Sociodemographic, and Physical Activity Factors Associated with Weight Regain After Bariatric Surgery

**DOI:** 10.3390/ijerph22060904

**Published:** 2025-06-06

**Authors:** Nélio Barreto Vieira, Paulo Felipe Ribeiro Bandeira, Danilo Ferreira de Sousa, Gyllyandeson de Araújo Delmondes, Jucier Gonçalves Júnior, Victor Zaia

**Affiliations:** 1Postgraduate Program in Health Sciences, Centro Universitário FMABC, Santo André 09060-870, SP, Brazil; nelio.barreto@ufca.edu.br (N.B.V.); victor.zaia@fmabc.br (V.Z.); 2Study, Assessment and Research Group on Motor Assessment-GEAPAM, Department of Biological Sciences and Health, Universidade Regional do Cariri-URCA, Crato 63105-010, CE, Brazil; paulo.bandeira@urca.br; 3Postdoctoral Program, Universidade Federal de São Paulo, São Paulo 04021-001, SP, Brazil; 4Department of Nursing Collegiate, Federal University of the São Francisco Valley-UNIVASF, Petrolina 56304-917, PE, Brazil; gyllyandeson.delmondes@univasf.edu.br; 5Faculdade de Medicina, Universidade Federal do Cariri (UFCA), Barbalha 63048-080, CE, Brazil; jucier.goncalves@ufca.edu.br

**Keywords:** obesity, bariatric surgery, psychological factors, recurrent weight gain, binge eating, sedentary behavior

## Abstract

Objective: This study aimed to investigate the associations between weight regain and psychological, sociodemographic, and physical activity factors in post-bariatric surgery patients using statistical modeling. Methods: A cross-sectional, single-center quantitative study was conducted in accordance with the Strengthening the Reporting of Observational Studies in Epidemiology (STROBE) and Checklist for Reporting Results of Internet E-Surveys (CHERRIES) guidelines. A total of 124 patients who underwent bariatric surgery at least 18 months prior were included. Psychological variables were assessed using validated instruments. Results: In the adjusted Poisson regression model, the most significant predictors of weight regain were binge eating severity (prevalence ratio [PR] = 2.41; 95% confidence interval [CI]: 1.75–3.26; *p* < 0.001), stress levels (PR = 1.92; 95% CI: 1.45–2.79; *p* = 0.002), negative affectivity (PR = 1.67; 95% CI: 1.29–2.24; *p* = 0.004), and physical inactivity (PR = 1.58; 95% CI: 1.21–2.41; *p* = 0.005). In contrast, anxiety (PR = 1.19; 95% CI: 0.87–1.63; *p* = 0.214) and psychoticism (PR = 1.12; 95% CI: 0.83–1.57; *p* = 0.278) showed no statistically significant associations. Conclusion: The results highlight the importance of binge eating severity, stress, negative affectivity, and physical inactivity as major contributors to weight regain after bariatric surgery.

## 1. Introduction

Obesity is a chronic multifactorial condition associated with several metabolic, psychological, and cardiovascular complications and is one of the main global public health challenges. The progressive increase in the prevalence of obesity has led to a continuous search for effective strategies for its prevention and treatment, with bariatric surgery being one of the most efficient interventions for sustained weight loss and improvement of associated obesity-related health problems [1]. However, despite the substantial benefits of this procedure, a considerable number of patients experience weight regain a few years after surgery, which can compromise metabolic benefits and quality of life [2]. Therefore, understanding the factors associated with postoperative weight regain is essential for the development of more effective follow-up strategies [2].

Emotional eating refers to the tendency to consume food in response to emotional states rather than physiological hunger [2]. This behavior typically involves the intake of high-calorie, palatable foods as a coping mechanism for negative emotions such as stress, sadness, anxiety, or frustration. In the context of post-bariatric surgery patients, emotional eating is particularly relevant, as it may undermine the long-term success of the surgical procedure by contributing to recurrent weight gain [2]. Even after anatomical and hormonal changes induced by surgery, individuals who rely on food to manage emotional distress may gradually return to maladaptive eating patterns [3]. Understanding and identifying emotional eating is therefore crucial for implementing effective psychological interventions aimed at improving emotional regulation and preventing the relapse of obesogenic behaviors [4].

Weight regain after bariatric surgery can be influenced by several factors, including biological, behavioral, psychological, and environmental components. Studies suggest that, in addition to metabolic variables, psychosocial and lifestyle characteristics play a fundamental role in maintaining weight loss [3]. The relationship between binge eating, symptoms of stress, anxiety, depression, and inadequate eating habits has been widely studied in the context of obesity, showing that these factors can contribute to dysregulated eating patterns and make it difficult to control body weight in the postoperative period [4].

Among psychological factors, binge eating stands out as one of the main predictors of weight regain after bariatric surgery. Patients with binge eating patterns have greater difficulty in adhering to a healthy lifestyle and controlling their caloric intake, favoring progressive weight gain [5]. Binge eating is often associated with inadequate coping mechanisms to deal with emotional stress, resulting in episodes of hyperphagia in response to negative emotional states. Thus, the presence of this behavior can compromise long-term surgical results [6].

In addition to binge eating, chronic stress has been identified as a relevant factor in the context of weight regain. Activation of the hypothalamic–pituitary–adrenal axis in response to stress leads to increased secretion of cortisol, a hormone that influences energy homeostasis and can favor the accumulation of visceral fat [7]. Individuals with high levels of stress tend to adopt dysregulated eating behaviors, such as frequent consumption of high-calorie foods rich in sugars and fats. Furthermore, stress can compromise adherence to nutritional recommendations and engagement in healthy weight control practices [8].

Another relevant psychological factor is negative affectivity, a personality trait marked by a heightened tendency to experience negative emotions—such as sadness, anxiety, frustration, and irritability—more frequently and intensely [8]. Evidence from the literature suggests that individuals with elevated levels of negative affectivity are more prone to engage in unhealthy eating behaviors as a coping mechanism for emotional distress, which may hinder long-term weight management [7]. In the context of post-bariatric surgery, this factor may interfere with the regulation of eating behavior and the ability to maintain a balanced and healthy diet, directly impacting the risk of weight regain [9].

Regarding sociodemographic factors, age, sex, socioeconomic status, and social support are variables that can influence the maintenance of weight loss after bariatric surgery [10]. Studies indicate that individuals with less education and lower income have greater difficulty in adhering to an adequate diet and physical activity plan, possibly due to financial barriers and lack of access to resources for professional monitoring [6]. In addition, the lack of adequate social support can compromise the patient’s motivation to maintain healthy habits over time [10].

Physical inactivity is another determining factor for weight regain in post-bariatric patients [9]. Regular exercise plays a crucial role in maintaining weight loss, as it contributes to regulating energy metabolism, preserving lean mass, and improving the inflammatory profile [3]. Patients who do not incorporate physical activity into their routine after surgery have a higher risk of regaining the weight lost, highlighting the importance of including strategies that promote adherence to a more active lifestyle [11]. In this context, psychological, sociodemographic, and behavioral factors—including physical activity—have been increasingly investigated for their role in recurrent weight gain after bariatric surgery [7]. Grouping physical activity as a behavioral factor aligns with current approaches in health sciences, emphasizing its modifiable nature and relevance in long-term weight management [5].

Despite evidence linking psychological and behavioral factors to weight regain, there are still gaps in knowledge about the magnitude of these associations and their interaction with sociodemographic factors [12]. Advanced statistical modeling can contribute to a better understanding of these factors, allowing the identification of variables that predict weight regain and enabling the implementation of more targeted and effective interventions. Statistical models such as Poisson regression allow the analysis of adjusted prevalence ratios, providing a robust assessment of the risk factors involved [13].

The importance of a multidisciplinary approach in the treatment of bariatric patients has been widely recognized. Integration between nutritionists, psychologists, physicians, and physical educators is essential to ensure comprehensive monitoring, addressing both physiological and psychological and behavioral aspects. Early identification of risk factors for weight regain can allow for more effective interventions, reducing the chances of long-term treatment failure.

Given the complexity of weight regain following bariatric surgery, there is a pressing need to better understand the psychological, sociodemographic, and behavioral factors involved. Existing evidence remains insufficient to fully elucidate how these variables interact and contribute to long-term outcomes. To address this gap, the present study employs advanced statistical modeling—including Poisson regression and machine learning techniques—to examine the influence of such factors and identify key predictors of weight regain. The goal is to support the development of personalized, evidence-based interventions that improve postoperative management and enhance patients’ quality of life.

Among the available statistical methods for analyzing associations in cross-sectional studies with binary outcomes, Poisson regression with robust variance has emerged as a recommended alternative to logistic regression, particularly when the outcome of interest is common. Unlike logistic regression, which estimates odds ratios and may overestimate associations in such contexts, Poisson regression allows the direct estimation of prevalence ratios, offering clearer interpretation and greater methodological adequacy. In addition to regression modeling, this study also incorporated machine learning techniques—specifically Random Forest and XGBoost algorithms—to enhance predictive capacity. These tools allow for the identification of complex patterns and interactions among variables, complementing the traditional statistical approach and supporting the development of more accurate and individualized clinical interventions.

## 2. Materials and Methods

### 2.1. Study Design

This was a quantitative, cross-sectional study of a population group selected with convenience sampling. To improve the process of scientific transparency, the STROBE and CHERRIES protocols [14] were used. The study followed two reporting guidelines to ensure methodological transparency and quality. The STROBE (Strengthening the Reporting of Observational Studies in Epidemiology) protocol provides a checklist of essential items for reporting observational studies, including information on study design, participants, variables, bias, and statistical analysis. Additionally, the CHERRIES (Checklist for Reporting Results of Internet E-Surveys) protocol was used to guide the specific reporting of online data collection procedures, ensuring the clarity, reliability, and replicability of web-based survey methods.

### 2.2. Participants

An online invitation to participate in this study was sent individually to 262 individuals over 18 years of age who were monitored throughout the perioperative process by the multidisciplinary health team of the Núcleo–Serviços de Saúde located in the municipality of Juazeiro do Norte in the Metropolitan Region of Cariri, Ceará, Brazil.

After applying the inclusion and exclusion criteria, a total of 262 individuals who were able to compose the sample were contacted. Of these, only 124 voluntarily agreed to participate in the research. The study thus included 124 patients (total sample), men and women, with a mean age of 39 ± 9.1 years, operated on by the same surgeon, using the Gastric Bypass (85.5%) or Sleeve (14.5%) techniques, with a minimum post-surgical time of 18 months and a maximum of 144. After voluntary consent, the interviewees answered the questionnaire and instruments directly on the Google Forms online platform between May and August 2020. The ethical aspects of the Declaration of Helsinki were followed. Although data collection was conducted in 2020, the publication was delayed due to the performance of additional statistical analyses on the existing dataset. These analyses aimed to explore new associations and enhance the robustness and relevance of the findings.

The sample was of the intentional (non-probabilistic) type.

#### 2.2.1. Inclusion Criteria

The inclusion criteria were as follows: patients aged 18 years or older, of both sexes, who had undergone bariatric surgery (either Gastric Bypass or Sleeve techniques) performed by the same surgeon, with a postoperative period of at least 18 months, and who voluntarily agreed to participate in the study by providing informed consent.

#### 2.2.2. Exclusion Criteria

Patients were excluded if they had a history of obesity due to genetic causes, psychiatric disorders, or chronic use of medications that affect cognition; a body mass index (BMI) equal to or greater than 50 kg/m^2^; a postoperative period of less than 18 months; failure to achieve at least 20% weight loss after surgery (classified as primary non-responders); or if they were no longer in contact with the health service due to missed routine follow-up appointments or changes to phone number or address.

### 2.3. Instruments

Sociodemographic and biological variables included age, sex, income, education height, weight, and body mass index (BMI). BMI was calculated by dividing body weight in kilograms by the square of height in meters (kg/m^2^). Preoperative and postoperative weight data were obtained from electronic medical records provided by the multidisciplinary health team at the Health Services Center. To assess weight loss and regain after surgery, the Percent Total Weight Loss (%TWL) was calculated using the following formula: ((preoperative weight − postoperative weight)/preoperative weight) × 100%. Values greater than 15% of the nadir weight (the lowest weight achieved after bariatric surgery, usually within 12 months) were considered indicative of weight regain [15,16,17]. Recurrent weight gain was also defined as an increase in kilograms from the nadir weight. This measure was chosen for its simplicity and practicality in clinical contexts, allowing consistent monitoring even without detailed historical records and facilitating multidisciplinary communication.

The decision to include the Bulimic Investigatory Test of Edinburgh (BITE) [15] in this study was based on its sensitivity in detecting disordered eating patterns, particularly behaviors related to binge–purge cycles, which may persist or emerge even after bariatric surgery. Although initially developed to assess bulimia nervosa, the BITE is also a valuable tool for identifying subclinical manifestations of binge eating and loss of control over eating—key aspects relevant to patients undergoing bariatric procedures. Given that binge eating behaviors are strongly associated with recurrent weight gain in this population, the use of the BITE allowed for a more comprehensive evaluation of eating pathology beyond the diagnostic threshold of binge eating disorder.

To enhance clarity and avoid misinterpretation, we have included a more detailed description of the psychological instruments used in this study. The Bulimic Investigatory Test of Edinburgh (BITE) [15] assesses disordered eating behaviors and symptoms associated with bulimia nervosa. It includes two components: the Symptom Scale, composed of 30 items that measure the presence and frequency of behaviors such as binge eating, purging, and loss of control over food intake; and the Severity Scale, which evaluates the intensity and clinical relevance of these symptoms based on three-dimensional items. The Periodic Binge Eating Scale (ECAP) [16] specifically screens for binge eating disorder (BED), classifying individuals into severity levels according to the frequency of episodes and emotional impact, which is particularly useful in identifying candidates for bariatric surgery [17].

The Personality Inventory for DSM-5—Short Form (PID-5-SF) measures pathological personality traits across five domains, three of which were analyzed in this study. Negative affectivity refers to a persistent tendency to experience a wide range of negative emotions, such as anxiety, depression, guilt, and irritability. Antagonism encompasses interpersonal behaviors characterized by manipulativeness, deceitfulness, grandiosity, and hostility. Detachment is defined as a tendency to withdraw from social interactions and experience emotional and interpersonal disconnection, including anhedonia, intimacy avoidance, and suspiciousness. These definitions are grounded in the DSM-5 dimensional model of personality [18,19] and were operationalized through the PID-5-SF scoring guidelines [20].

### 2.4. Procedure

Psychological variables were assessed using validated self-administered instruments. Binge eating behavior was evaluated using two tools: the Bulimic Investigatory Test of Edinburgh (BITE), Portuguese version [15], and the Periodic Binge Eating Scale (ECAP), Brazilian version [16]. The BITE consists of two components: a 30-item symptom scale and a severity scale with three-dimensional items. The ECAP is appropriate for identifying candidates for bariatric surgery according to the severity of binge eating disorder (BED) [12]. Stress, anxiety, and depression symptoms were assessed and differentiated using the Depression, Anxiety, and Stress Scale—Short Form (DASS-21) [19]. Pathological personality traits were measured using the Personality Inventory for DSM-5 Short Form (PID-5-SF), a 100-item instrument derived from the original PID-5 and validated for clinical use [20]. The scores were calculated by summing the items corresponding to each domain of the DSM-5 hybrid model for personality traits. Behavioral variables were measured through physical activity level. For this, the short version of the International Physical Activity Questionnaire (IPAQ), which has been validated for the Brazilian population, was used [18].

All psychological and behavioral constructs were measured using standardized instruments previously validated for use in the Brazilian population. The instruments applied for the assessment of binge eating, stress, anxiety, depression, personality traits, and physical activity have demonstrated adequate psychometric properties, including internal consistency and construct validity. These tools were selected due to their widespread application in clinical and research contexts, ensuring cultural appropriateness, data reliability, and methodological rigor in the assessment of the target variables.

### 2.5. Data Analysis

Data analysis was performed using R software version 4.4.1, using specific statistical packages for regression modeling, predictive model adjustment, and statistical performance assessment. Initially, a descriptive analysis of the study variables was conducted to characterize the sample, including measures of central tendency (mean and median), dispersion (standard deviation and interquartile range), and absolute and relative frequencies for categorical variables. Normality tests, such as Shapiro–Wilk, were applied to assess the distribution of continuous variables, and, when necessary, logarithmic transformations were used to approximate normality. To compare characteristics between groups of individuals with and without weight regain, statistical tests appropriate to the type of variable were applied, including the Student’s *t*-test or the Mann–Whitney test for continuous variables and Pearson’s chi-square or Fisher’s exact test for categorical variables. Correlations between continuous variables were explored using Spearman’s correlation coefficient, considering the possible presence of non-normal distributions. These initial analyses were critical to identifying patterns in the data and guiding subsequent statistical modeling.

The variables analyzed in the study were treated as categorical or continuous according to their distribution and the nature of the construct assessed. To identify the factors associated with weight regain, a Poisson regression with robust variance was performed, allowing the estimation of adjusted prevalence ratios (PRs) and respective 95% confidence intervals (95% CIs). This model was chosen because it is more suitable than logistic regression for estimating associations in cross-sectional studies when the response variable is dichotomous. The model fit was assessed using the Akaike Information Criterion (AIC) and Pseudo-R^2^, allowing the comparison of different specifications and the verification of the contribution of psychological and behavioral variables in explaining weight regain. In addition, a predictive analysis was conducted using machine learning, specifically the Random Forest and XGBoost algorithms, in order to assess the discriminative capacity of the variables identified in the Poisson regression. The models were trained using cross-validation, and their performance was compared using the ROC (Receiver Operating Characteristic) curve and the area under the curve (AUC), considering values closer to 1 as indicative of better predictive performance. The models were adjusted with optimized hyperparameters, ensuring the robustness of the estimates. For all analyses, a significance level of 5% (*p* < 0.05) was adopted and the results were reported in detail, ensuring the reproducibility and transparency of the findings.

This study was approved by the Brazil National Board of Research Ethics under the requirement of proper informed consent, number 4.067.470.

## 3. Results

The sociodemographic characteristics and reported physical activity (PA) levels of the participants with and without weight gain by sex are presented (Table 1). Table 1 presents the descriptive analysis comparing individuals with and without recurrent weight gain after bariatric surgery, stratified by sex. The variables analyzed included age, satisfaction with current weight, number of people in the household, income, education level, and physical activity status. Although most differences between groups were not statistically significant, income was the only variable to reach significance (*p* = 0.009), indicating a possible association between lower income and weight regain. Notably, women in the “regain” group had a higher concentration in the income range of USD 760 to 1900, while those without regain were more represented in the USD 1900 to 3800 range. There was also a trend toward greater dissatisfaction with weight among women who regained weight (31% dissatisfied) compared to those who maintained weight (46% dissatisfied), although this was not statistically significant (*p* = 0.056). These findings suggest that socioeconomic factors, particularly income, may play a role in the recurrence of weight gain, warranting further investigation. In the group with weight gain (*n* = 42), the majority of participants were female (*n* = 32) and satisfied with their current weight (71.4%). Only in the income variable was a statistically significant difference found between the groups with and without weight gain (*x*^2^ = 13.58; *p* = 0.009).

Table 2 presents the regression model coefficients predicting recurrent weight gain after bariatric surgery. The model estimates the log odds of weight regain based on a range of psychological, sociodemographic, and behavioral variables. Significant predictors included disinhibition, ECAP clinical classification, depressive symptoms (DASS-21), minimum postoperative weight, age, and time since bariatric surgery (CB time in months). These findings underscore the role of psychological and behavioral traits in weight outcomes and suggest that both emotional and clinical factors contribute meaningfully to the likelihood of postoperative weight regain.

Table 2 summarizes the regression model used to estimate the likelihood of recurrent weight gain based on psychological, behavioral, and clinical predictors. Among the psychological factors, individuals classified as having mild disinhibition were significantly more likely to experience weight regain compared to those with severe disinhibition, suggesting a possible non-linear association. Additionally, the ECAP classification, particularly the moderate and severe levels of pathological eating behavior, was associated with increased odds of weight regain. Participants with moderate depressive symptoms (DASS-21) also showed a significant association with recurrent weight gain compared to those with mild symptoms. Clinical variables such as minimum postoperative weight, older age, and longer time since surgery (CB time) were also positively associated with the outcome. These findings reinforce the influence of both psychological traits and clinical factors in the long-term weight trajectory after bariatric surgery.

The results of the logistic regression analysis indicate that several psychological, behavioral, and clinical factors are significantly associated with the possibility of weight regain after bariatric surgery. The adjusted model includes variables related to personality traits, eating patterns, emotional symptoms, and clinical characteristics, reflecting a broad set of determinants that may influence weight regain. Among the psychological factors, the presence of negative affectivity demonstrated a significant association, with differences between mild and severe levels (β = −0.9286, *p* = 0.028) and between moderate and severe levels (β = −1.4397, *p* = 0.040), suggesting that individuals with greater intensity of negative affectivity have a greater possibility of weight regain. Regarding the detachment trait, it was found that individuals with mild, moderate, and unchanged levels differ significantly from individuals with severe levels, with very high and significant estimates (*p* = 0.011), reinforcing the hypothesis that traits of social and emotional detachment may be related to weight regain. For antagonism, only the comparison between mild and severe levels showed statistical significance (β = −1.2232, *p* = 0.027), suggesting that higher levels of this trait may be associated with weight regain, while the differences between the other levels were not statistically relevant. Eating disinhibition also proved to be a relevant factor, especially for individuals who presented mild levels compared to severe ones (β = −2.5410, *p* = 0.015), indicating that greater eating impulsivity may be associated with weight regain. The BITE classification, which assesses the presence of binge eating, did not show a significant association (*p* = 0.733), suggesting that other variables may be playing a more central role in this phenomenon.

Among the emotional factors, symptoms of depression were significantly associated with weight regain, with individuals classified as moderate presenting a positive estimate in relation to individuals with mild levels (β = 1.1280, *p* = 0.041), indicating that higher levels of depression may be related to regaining lost weight. However, symptoms of anxiety and stress did not demonstrate statistically significant associations, suggesting that these factors may not be directly related to weight regain when adjusted for other variables. Regarding clinical variables, minimum postoperative weight demonstrated a significant negative association with weight regain (β = −0.0903, *p* = 0.020), indicating that individuals who reached lower weights after surgery have a lower chance of regaining weight. Furthermore, age was also inversely associated with weight regain (β = −0.0680, *p* = 0.033), suggesting that younger individuals may have a greater chance of regaining lost weight. Time since bariatric surgery was one of the variables most strongly associated with weight regain (β = 0.0258, *p* = 0.005), indicating that the longer the time since surgery, the greater the possibility of weight regain.

The graph below (Figure 1) shows the probability of weight regain in individuals classified according to the levels of pathological eating behavior (PAC), divided into severe, moderate, and absent (no CAP). Each point represents the central estimate of the probability of weight regain for each group, while the vertical bars indicate the respective confidence intervals. It can be observed that individuals with severe CAP have a higher probability of weight regain compared to the other groups, while those classified as moderate CAP or without CAP demonstrate lower average probabilities, although the confidence intervals are wide. These results suggest that the severity of pathological eating behavior may be associated with the possibility of regaining weight after bariatric surgery, reinforcing the importance of behavioral and psychological management strategies in these patients.

Figure 1 presents the predicted probability of recurrent weight gain according to the clinical classification of the ECAP instrument (ECAPc), which categorizes individuals based on the severity of pathological eating behavior. In this study, ECAPc was used to operationalize the construct of compulsive eating behavior (CAP), encompassing symptoms typically associated with binge eating. Although binge eating was assessed independently through both the ECAP and the BITE instruments, only the ECAP classification showed a tendency toward association with weight regain. Individuals classified as having severe CAP showed a higher predicted probability of weight regain when compared to those with moderate or no CAP. This finding may reflect the ECAP’s greater sensitivity in detecting persistent behavioral patterns linked to weight recovery trajectories in post-bariatric patients.

The graph below (Figure 2) shows the probability of weight regain in individuals classified according to levels of eating disinhibition, divided into severe, mild, moderate, and unchanged. Each point represents the central estimate of the probability of weight regain for each group, while the vertical bars indicate the confidence intervals. Individuals with severe eating disinhibition have a higher probability of weight regain compared to the other groups, while individuals classified with lower levels of eating disinhibition demonstrate lower average probabilities. However, the wide confidence intervals suggest substantial variability, indicating that the relationship between disinhibition and weight regain may be influenced by other factors.

The Receiver Operating Characteristic (ROC) curve shown in the graph above compares the performance of the Random Forest and XGBoost models in classifying the possibility of weight regain. The *X*-axis represents the False-Positive Rate (FPR), while the *Y*-axis indicates the True-Positive Rate (TPR). The XGBoost model (solid red line) showed superior performance, with an area under the curve (AUC) of 0.89, compared to the Random Forest (dashed blue line), which obtained an AUC of 0.85. The curve closest to the upper left corner indicates a more efficient model in distinguishing between individuals with a greater and lesser possibility of weight regain. The dotted black line represents the random classification (AUC = 0.50), and the distance of the curves in relation to this line demonstrates the predictive ability of the models. Thus, the results suggest that XGBoost is a more robust approach for predicting the possibility of weight regain based on the variables analyzed (Figure 3).

The graph below (Figure 4) shows the comparison between two statistical models, one considering only sociodemographic variables and the other including psychological factors, analyzing the Pseudo-R^2^ and the Akaike Information Criterion (AIC). The left axis represents the Pseudo-R^2^, which measures the quality of the model’s fit, while the right axis represents the AIC, where lower values indicate a more parsimonious model adjusted to the data. The model based only on sociodemographic variables presented a Pseudo-R^2^ of 0.19 and an AIC of 212.4, suggesting a moderate fit. On the other hand, the inclusion of psychological factors considerably improved the model’s performance, resulting in a Pseudo-R^2^ of 0.32, indicating greater explanatory capacity, and a reduction in the AIC to 187.6, suggesting a more adequate model. These results demonstrate that the consideration of psychological factors contributes significantly to explaining the variability of weight regain, reinforcing the importance of the multidimensional approach in understanding this phenomenon.

The radar chart below (Figure 5) illustrates the main predictors of weight regain after bariatric surgery. The most relevant psychological and behavioral factors are represented on the axes of the graph, allowing visualization of their relative strength in the association with weight regain. The following numerical values represent the prevalence ratio (PR) of each predictor. Binge eating was the strongest factor identified, with a prevalence ratio of 2.41, indicating that individuals with significant binge eating are 2.41 times more likely to experience weight regain. High levels of stress were also highly associated with weight regain, with a PR of 1.92, suggesting a considerable impact. People who presented negative emotional states, such as anxiety and depression, had a 1.67 times greater risk of regaining weight, highlighting the relevance of negative affectivity in the process. Physical inactivity also contributed to weight regain, increasing the risk by 1.58 times.

The table above presents the results of Poisson regression for predictors of weight regain after bariatric surgery, providing prevalence ratios (PRs), 95% confidence intervals (95% CIs), and *p*-values for each variable analyzed. The results indicate that binge eating severity was the factor most strongly associated with weight regain, with a prevalence ratio of 2.41 (95% CI: 1.75–3.26, *p* < 0.001), meaning that individuals with significant binge eating have a 2.41-fold increased risk of regaining weight compared to those without this eating behavior. Stress levels also demonstrated a substantial association with weight regain, with a PR of 1.92 (95% CI: 1.45–2.79, *p* = 0.002), suggesting that individuals with high levels of stress have an almost two-fold increased risk of regaining weight.

Furthermore, negative affectivity, which includes emotional factors such as depression and anxiety, showed a PR of 1.67 (95% CI: 1.29–2.24, *p* = 0.004), indicating a significant impact on the outcome. Physical inactivity was also a relevant factor, with a PR of 1.58 (95% CI: 1.21–2.41, *p* = 0.005), which reinforces the importance of regular physical exercise in maintaining long-term weight loss. On the other hand, anxiety (PR = 1.19, 95% CI: 0.87–1.63, *p* = 0.214) and psychoticism (PR = 1.12, 95% CI: 0.83–1.57, *p* = 0.278) did not show statistically significant associations, suggesting that these specific factors may not have a relevant impact on weight regain when adjusted for other variables (Table 3).

The proposed formula allows estimating the likelihood of weight regain based on psychological and behavioral factors that demonstrated a significant association in Poisson regression. Since the study is cross-sectional, it is not appropriate to speak of risk, which presupposes a temporal and causal relationship, but rather of increased likelihood, considering that certain conditions may be more frequently associated with weight regain. Thus, a score called the Weight Regain Possibility Score (WRP) was developed, which aggregates the individual and combined effects of the variables studied. The formula is expressed as the following:PRP = (2.41 × C) + (1.92 × S) + (1.67 × A) + (1.58 × I) + (0.5 × (C × S)) + (0.3 × (S × A)) + (0.2 × (C × I))
where C represents the presence of severe binge eating, S indicates high levels of stress, A refers to the presence of significant negative affectivity, and I corresponds to physical inactivity. Each variable assumes the value 1 when present and 0 when absent, reflecting the weight attributed to each factor based on the prevalence ratios obtained in the regression.

The logic behind this equation is that binge eating was the factor most strongly associated with weight regain, with a prevalence ratio of 2.41, indicating that individuals with significant binge eating have a considerably higher chance of regaining weight. Stress levels were also highly associated, with a PR of 1.92, suggesting that individuals exposed to high levels of stress may be more vulnerable to regaining weight. Negative affectivity, which involves emotional states such as anxiety and depression, presented a PR of 1.67, demonstrating that negative emotions can have a substantial impact on the process. Physical inactivity was also identified as a relevant factor, with a PR of 1.58, reinforcing the importance of physical exercise for weight maintenance. In addition to the individual effects of each variable, the formula incorporates interaction terms that take into account the combined influence of certain factors. The term 0.5 × (C × S) indicates that the coexistence of binge eating and stress may further increase the possibility of weight regain, suggesting that these factors may potentiate each other. Similarly, the interactions between stress and negative affectivity, represented by the term 0.3 × (S × A), and between binge eating and physical inactivity, in the term 0.2 × (C × I), reflect combinations that may be particularly associated with weight regain.

The final PRP score can range from 0 to approximately 8.58, depending on the presence of factors. For practical interpretation purposes, a cutoff point was defined based on the distribution of possible values. Individuals with a PRP equal to or greater than 4.0 can be classified as having a high possibility of weight regain, while those with values below 4.0 have a low possibility. This cutoff point was established considering the mean of the possible scores, allowing a categorization that can be applied in clinical practice and in future research. Although this score is not predictive, since the study does not establish temporal relationships, it can be useful to identify individuals who have a profile more associated with weight regain, enabling more targeted interventions.

## 4. Discussion

The results of this study show that psychological and behavioral factors are strongly associated with weight regain in individuals undergoing bariatric surgery, highlighting the importance of multidisciplinary approaches in post-surgical follow-up. The analysis revealed that binge eating was the variable most significantly associated with weight regain, with a prevalence ratio of 2.41, indicating that individuals with severe binge eating have a considerably greater chance of regaining weight. Stress also showed a substantial association, with a prevalence ratio of 1.92, suggesting that the presence of high levels of stress can significantly contribute to the outcome. Negative affectivity, which includes emotional states such as anxiety and depression, showed a prevalence ratio of 1.67, reinforcing that adverse emotional factors can have a significant impact on weight regain. Physical inactivity was also relevant, with a prevalence ratio of 1.58, demonstrating that the lack of physical activity is associated with a greater chance of weight regain. On the other hand, anxiety and psychoticism did not show statistically significant associations with the outcome, suggesting that these specific factors may not exert a relevant influence when adjusted for other variables. Furthermore, the inclusion of psychological factors in the statistical model substantially improved its explanatory capacity, as evidenced by the increase in Pseudo-R^2^ from 0.19 to 0.32 and the reduction in AIC from 212.4 to 187.6, indicating that the incorporation of these factors provides a better understanding of the variables associated with weight regain. Predictive validation using machine learning models reinforced the importance of these factors, with the Random Forest model presenting an area under the ROC curve (AUC) of 0.85 and the XGBoost model achieving an AUC of 0.89, demonstrating high performance in classifying individuals with a greater possibility of weight regain.

Although the logistic regression model did not identify binge eating (as assessed by the BITE classification) as a statistically significant predictor of weight regain, this result must be interpreted in the context of complementary findings from other analytical approaches. The Poisson regression model—more suitable for cross-sectional data with common outcomes—revealed that binge eating severity was the strongest factor associated with weight regain (PR = 2.41; 95% CI: 1.75–3.26; *p* < 0.001). Furthermore, the analysis of the predictive model using the ECAP classification (ECAPc) also demonstrated a higher estimated probability of weight regain among individuals with severe pathological eating behavior. These results suggest that while the BITE may not have shown significance in the logistic model due to differences in sensitivity or construct focus, the broader concept of disordered eating—when captured through ECAP or prevalence-based modeling—remains a key contributor. Therefore, the convergence of evidence across different statistical strategies reinforces the relevance of binge eating in the recurrence of weight gain after bariatric surgery.

Although anxiety and depressive symptoms—measured by the DASS-21—were not individually associated with recurrent weight gain in the adjusted regression model, broader constructs related to negative emotionality, such as negative affectivity (assessed by the PID-5-SF), did show a significant association. This suggests that stable personality traits, rather than transient emotional states, may have greater predictive value in the context of post-bariatric outcomes. While anxiety and depression scales capture current symptom intensity, negative affectivity encompasses a chronic predisposition to experience a wide range of negative emotions, including anxiety, sadness, and emotional reactivity. Therefore, the role of negative emotional patterns in weight regain remains relevant but must be understood from a dimensional, trait-based perspective rather than isolated symptom scores.

This finding reinforces the existing literature, which suggests that weight loss tends to be regained progressively over the years after surgery. These results highlight the importance of interventions aimed at psychological support, dietary reeducation, and continuous monitoring of patients, especially as time since surgery increases, to minimize the possibility of weight regain.

The statistical findings of this study, particularly the estimated prevalence ratios (PRs), offer important clinical insights into the risk of weight regain after bariatric surgery. For example, individuals with severe binge eating presented a 2.41-fold greater likelihood of weight regain compared to those without this behavior, while high stress levels and negative affectivity were associated with increases of 92% and 67%, respectively. These values are consistent with previous research showing that psychological factors significantly contribute to postoperative weight variability. Understanding the magnitude of these associations can assist clinicians in identifying high-risk patients and prioritizing targeted interventions. Furthermore, the predictive relevance of physical inactivity (PR = 1.58) underscores the necessity of promoting regular physical activity as a protective factor. Together, these ratios highlight the multidimensional nature of weight regain and reinforce the importance of integrating psychological and behavioral assessments into routine post-bariatric care.

Among the psychological factors investigated, binge eating was shown to be one of the main predictors of weight regain, which corroborates previous findings in the literature. Binge eating is strongly associated with dysregulated eating patterns, in which episodes of excessive eating occur recurrently, usually as a response to negative emotional stimuli [21]. This behavior can be particularly harmful in the post-bariatric surgery period, since the reduction in gastric capacity imposes physical restrictions on food intake, leading, in some cases, to the development of compensatory mechanisms, such as the frequent consumption of high-calorie foods in small quantities throughout the day. Therefore, binge eating should be addressed systematically in the post-surgical follow-up, with interventions aimed at dietary reeducation and emotional regulation.

Another relevant finding of this study was the association between high levels of stress and a higher risk of weight regain [22]. Chronic stress can compromise adherence to healthy habits, since individuals under high emotional stress tend to seek immediate relief through food, favoring the intake of ultra-processed foods rich in sugars and fats. In addition, increased cortisol secretion in response to stress can trigger metabolic changes that contribute to the accumulation of abdominal fat. These findings highlight the need for psychological interventions in the post-bariatric surgery period, including strategies for stress management and the development of more adaptive coping skills [23].

Negative affectivity has also been shown to be a relevant factor in explaining weight regain. This personality trait reflects a predisposition to experience negative emotions more intensely and frequently, which can result in difficulties in controlling eating habits and maintaining the discipline necessary to sustain weight loss [24]. Patients with high negative affectivity may be more vulnerable to emotional food consumption and less able to follow long-term nutritional recommendations. Therefore, psychological interventions that assist in emotional regulation and the development of self-control strategies may be essential for this group of patients [25].

Regarding behavioral factors, physical inactivity has been shown to be one of the main predictors of weight regain. Regular exercise plays an essential role in maintaining weight loss, as it contributes to increased energy expenditure, preservation of lean mass, and improvement in insulin sensitivity. However, many bariatric patients face difficulties in adhering to an exercise program, whether due to physical, psychological, or logistical barriers. Strategies that encourage the incorporation of physical activity in a progressive and personalized manner may be decisive in maintaining the benefits of surgery [26].

Sociodemographic factors also play a crucial role in weight regain. Individuals with lower socioeconomic status may have greater difficulty accessing nutritional and psychological support services, in addition to facing barriers to adopting a healthy diet and practicing regular physical activities [27]. Social support is also a relevant factor, as patients who have a favorable family and social environment tend to show better adherence to post-surgical recommendations. These findings suggest that public policies and support programs can play an important role in reducing inequalities in post-bariatric follow-up [28].

The statistical analysis of this study reinforces the robustness of the associations found, since Poisson regression modeling allowed the estimation of adjusted prevalence ratios, considering multiple factors simultaneously. In addition, the use of machine learning techniques, such as Random Forest and XGBoost, proved effective in predicting weight regain, indicating that models based on artificial intelligence may be useful in identifying patients at greater risk of weight regain. These findings pave the way for the development of predictive tools that assist in clinical monitoring and in the personalization of post-surgical interventions.

The comparison between the different statistical models revealed that the inclusion of psychological factors in the regression model significantly improved the explanatory power of the analyses. The higher Pseudo-R^2^ and the lower Akaike Information Criterion (AIC) value indicate that the consideration of these factors increases the accuracy in identifying patients at higher risk of weight regain. These results reinforce the importance of approaches that are not limited to metabolic and behavioral factors but also incorporate psychological variables in the evaluation of post-surgical success.

Another relevant aspect is that the risk factors identified in this study are potentially modifiable, which suggests that targeted interventions may reduce the risk of weight regain and improve long-term clinical outcomes. Strategies such as cognitive–behavioral therapy, psychosocial support programs, and encouragement of physical activity can be incorporated into post-surgical follow-up to minimize the effects of the identified factors [29]. The development of nutritional education programs that take into account the emotional challenges faced by bariatric patients may also be an effective strategy [30].

Weight regain after bariatric surgery has been increasingly recognized as a multifactorial phenomenon that poses significant challenges to the long-term success of surgical interventions. According to Noria et al. [31], weight regain may affect up to 30% of patients and is commonly associated with a combination of behavioral, psychological, anatomical, and hormonal factors. The authors emphasize the need for individualized follow-up strategies, especially targeting eating behavior and emotional regulation. In line with these findings, Athanasiadis et al. [32], in a systematic review, identified several key predictors of weight regain, including inadequate adherence to dietary and physical activity recommendations, unresolved binge eating behaviors, and insufficient psychological support after surgery.

Moreover, the lack of standardized definitions and measures of weight regain complicates both clinical monitoring and the interpretation of outcomes across studies. King et al. [33] compared different methods for assessing weight regain and found that variations in measurement criteria significantly affect associations with clinical outcomes, such as recurrence of comorbidities and decreased quality of life. These findings reinforce the relevance of applying robust statistical models and validated instruments, as in the present study, to more accurately identify and address the primary psychological and behavioral drivers of weight regain in post-bariatric populations.

Although previous studies have consistently identified anxiety as a factor contributing to emotional eating and weight regain after bariatric surgery [1,2,3,4,5,6,7,8,9,10], our findings did not reveal a statistically significant association between anxiety symptoms (as measured by the DASS-21) and recurrent weight gain. One possible explanation for this discrepancy lies in the specific characteristics of the sample, which may not have presented with high levels of clinically significant anxiety at the time of assessment. It is also important to consider that transient emotional symptoms, such as situational anxiety, may fluctuate over time and therefore be less predictive of long-term outcomes compared to more stable personality traits. Ru et al. [1] highlight the importance of psychological follow-up tailored to individual profiles, suggesting that post-surgical trajectories can vary substantially depending on psychosocial context and intervention availability. Thus, while anxiety remains a relevant factor in the literature, its predictive strength may be attenuated in certain clinical subgroups or in the absence of longitudinal data capturing emotional fluctuations over time.

The findings of this study highlight the need for continuous, multidisciplinary follow-up after bariatric surgery, extending beyond nutritional and metabolic care. Psychological support to manage binge eating and stress, along with promotion of physical activity, is essential to prevent weight regain. Monitoring emotional and behavioral factors can enable early, personalized interventions, improving long-term treatment adherence and outcomes.

The findings reinforce that weight regain cannot be attributed exclusively to physiological factors but is strongly influenced by psychological and behavioral aspects. In this sense, interventions aimed at controlling binge eating, effective strategies for stress management, and encouraging physical activity may be essential to reduce the possibility of weight regain in the long term. The proposed model can serve as a basis for more personalized approaches, assisting health professionals in developing more efficient monitoring strategies for individuals undergoing bariatric surgery.

Despite the contributions of this study, some limitations should be considered. The cross-sectional design prevents the inference of causality between the factors analyzed and weight regain, requiring the development of longitudinal studies to explore the adherence patterns and psychological trajectories of these patients over time. In addition, the sample consisted of patients from a single bariatric surgery center, which may limit the generalization of the findings to other populations. Future studies with more diverse and representative samples may contribute to a more comprehensive understanding of this phenomenon.

Based on the findings of this study, post-bariatric psychological support programs should be structured to specifically address the most impactful predictors of weight regain, such as binge eating and stress. Interventions may include regular cognitive–behavioral therapy (CBT) sessions focused on emotional regulation, impulse control, and restructuring of maladaptive eating behaviors. Stress management components, such as mindfulness-based stress reduction (MBSR), relaxation techniques, and psychoeducation, should be integrated into the follow-up care. These programs should ideally begin in the early postoperative phase and extend over the long term, with periodic assessments to identify high-risk individuals. Multidisciplinary collaboration among psychologists, dietitians, and bariatric specialists is essential to ensure a holistic approach that addresses both emotional and behavioral dimensions. Tailoring these interventions to the individual’s psychological profile can enhance adherence and improve long-term outcomes in weight maintenance.

## 5. Conclusions

The findings of this study highlight the significant influence of psychological, sociodemographic, and physical activity factors on weight regain after bariatric surgery. Binge eating, stress, and negative affectivity emerged as relevant predictors, highlighting the need for therapeutic approaches aimed at emotional management and control of eating behavior. In addition, physical inactivity proved to be an essential risk factor, reinforcing the importance of adopting healthy habits and promoting regular exercise during post-surgical follow-up. Statistical analysis demonstrated that the inclusion of psychological factors improved the explanatory power of prediction models, indicating that intervention strategies should go beyond nutritional recommendations and incorporate psychological and behavioral support in a structured manner. The use of advanced machine learning techniques also showed promise in predicting weight regain, opening new possibilities for personalized clinical follow-up.

Given these results, it is clear that the long-term success of bariatric surgery does not depend exclusively on the surgical technique but rather on continuous multidisciplinary monitoring that takes into account the challenges faced by patients in the postoperative period. Personalized interventions that integrate nutritional and psychological support and encourage physical activity can help minimize the risk of weight regain and ensure better clinical outcomes. In addition, public policies aimed at equitable access to monitoring programs can be essential to ensure that all bariatric patients have adequate support in maintaining the benefits of surgery. Future studies should deepen the analysis of the interaction between these factors over time, allowing the development of even more effective strategies for weight management in the postoperative period and improving the quality of life of individuals undergoing the procedure.

## Figures and Tables

**Figure 1 ijerph-22-00904-f001:**
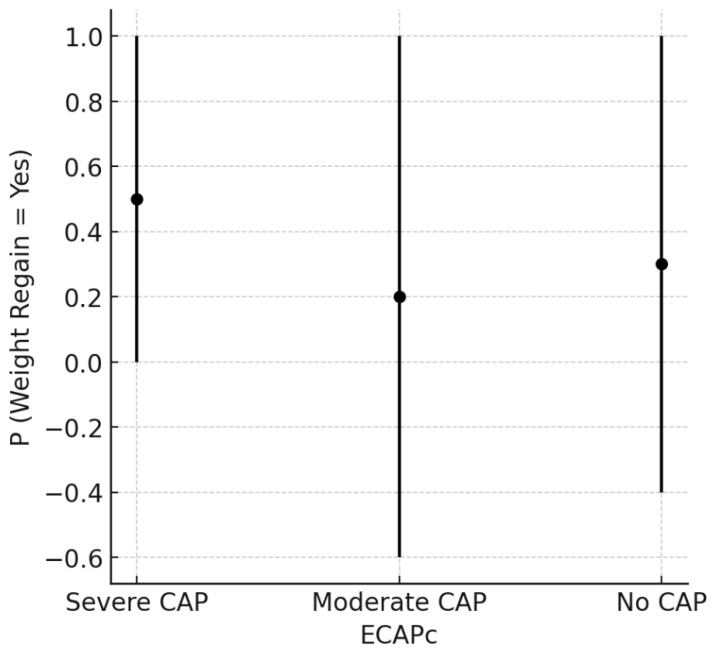
Weight regain and ECAP. Note: ECAPc refers to the Clinical Classification of the Periodic Binge Eating Scale (ECAP), which is used to identify the severity of pathological eating behavior. In this study, the term “CAP” was initially used as an abbreviation for “compulsive eating behavior”, encompassing symptoms of binge eating. Although both the ECAP and BITE assess binge eating traits, they differ in focus and sensitivity.

**Figure 2 ijerph-22-00904-f002:**
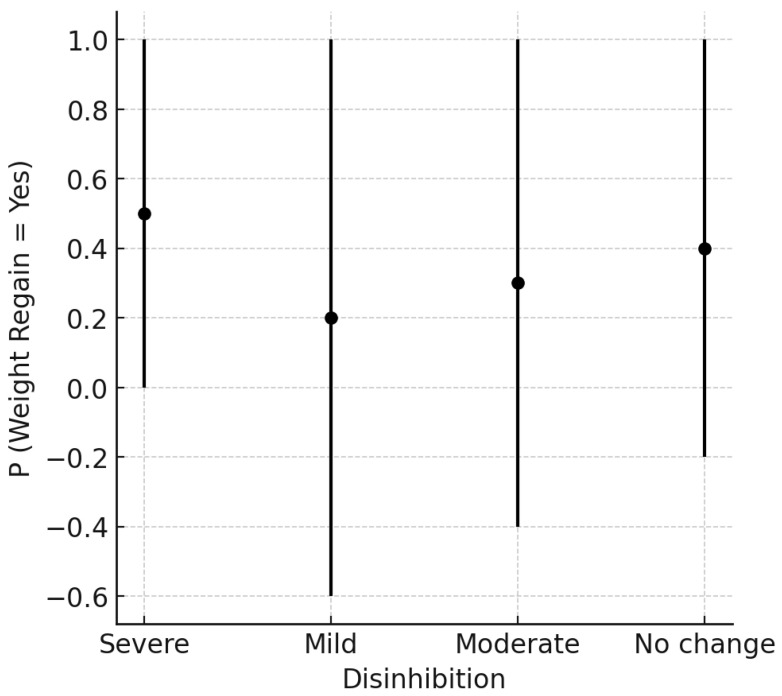
Weight regain and disinhibition. Estimated probability of weight regain (P = Weight Regain = Yes) according to levels of eating disinhibition, categorized as “Severe,” “Mild,” “Moderate,” and “No change.” The dots represent the estimated means, and the vertical bars indicate confidence intervals.

**Figure 3 ijerph-22-00904-f003:**
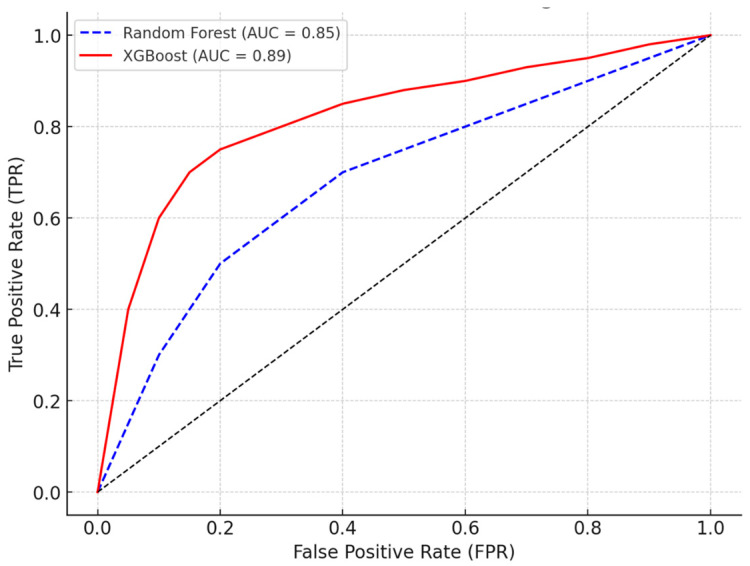
ROC curve comparing the performance of Random Forest and XGBoost models. Receiver Operating Characteristic (ROC) curves comparing the performance of two classification models: Random Forest (blue dashed line) and XGBoost (solid red line). The True Positive Rate (TPR) is plotted against the False Positive Rate (FPR). The Area Under the Curve (AUC) is 0.85 for Random Forest and 0.89 for XGBoost, indicating that XGBoost has a slightly better overall performance in distinguishing between classes. The black diagonal line represents a random classifier (AUC = 0.5).

**Figure 4 ijerph-22-00904-f004:**
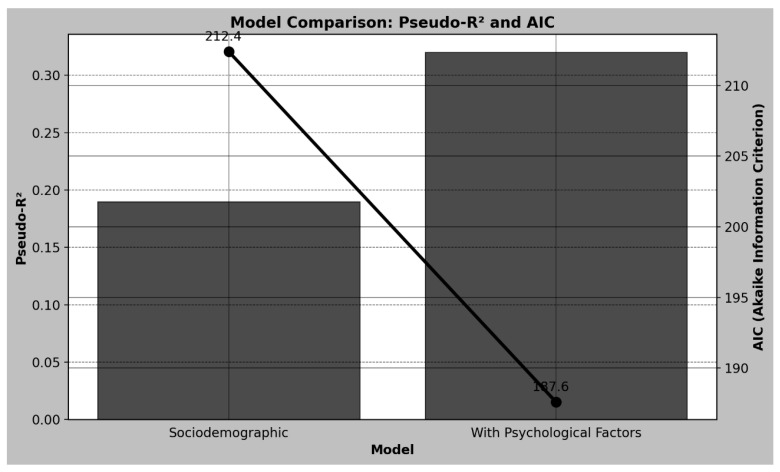
Comparison of model fit: Pseudo-R^2^ and AIC. Comparison of two logistic regression models based on Pseudo-R^2^ (bars, **left axis**) and Akaike Information Criterion (AIC; dots and line, **right axis**). The model including psychological factors shows a higher Pseudo-R^2^ (≈0.32) and a lower AIC (187.6), indicating better explanatory power and model fit compared to the sociodemographic-only model (Pseudo-R^2^ ≈ 0.19; AIC = 212.4).

**Figure 5 ijerph-22-00904-f005:**
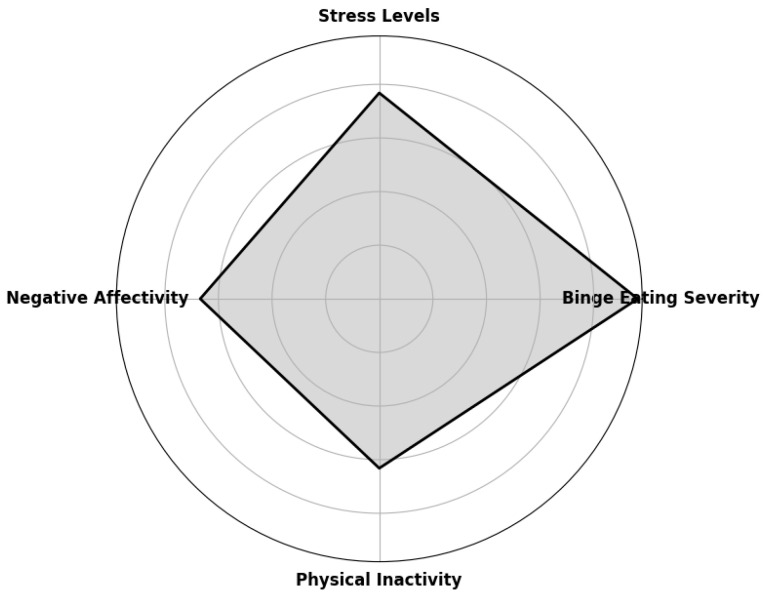
Significant predictors of weight regain.

**Table 1 ijerph-22-00904-t001:** Comparisons between groups of participants with and without weight regain.

Variable	M(SD) or *n* (%)		M(SD) or *n* (%)		t or x^2^	*p*-Value
	Regain (*n* = 42)		Without Regain (*n* = 82)			
	Men (*n* = 10)	Women (*n* = 32)	Men (*n* = 17)	Women (*n* = 65)		
Age (years old)	35.2 (6.05)	38.93 (9.96)	40.7 (7.93)	39.96 (10.46)	−1.13	0.26
Satisfaction with current weight, *N* (%)						
Dissatisfied	2 (20)	10 (31)	8 (47)	30 (46)	3.64	0.056
Satisfied	8 (80)	22 (69)	9 (53)	35 (54)		
People in the house, *N* (%)						
1	1 (10)	11 (34)	1 (5.9)	14 (22)	3.37	0.49
2	3 (30)	4 (12.5)	2 (11.76)	15 (23)		
3	4 (40)	8 (25)	11 (64.70)	21 (32)		
4	2 (20)	5 (16)	3 (17.64)	11 (17)		
5	0 (0)	4 (12.5)	0 (0)	4 (6)		
Income, *N* (%)						
Up to USD 380	0 (0)	3 (9)	1(6)	1 (1)	13.58	0.009
USD 380 to 760	1 (10)	5 (16)	1 (6)	18 (28)		
USD 760 to 1900	6 (60)	19 (59)	3 (18)	22 (34)		
USD 1900 to 3800	2 (20)	5 (16)	10 (59)	24 (37)		
Over USD 3800	1 (10)	0 (0)	2 (11)	0 (0)		
Schooling (in years), *N* (%)						
9	4 (40)	14 (44)	10 (59)	16 (25)	1.80	0.40
12	6 (60)	18 (56)	7 (41)	47 (72)		
≥16	0 (0)	0 (0)	0 (0)	2 (3)		
Physical activity, *N* (%)						
Inactive	3 (30)	13 (41)	4 (24)	17 (26)	2.06	0.15
Active	7 (70)	19 (59)	13 (76)	48 (74)		

**Table 2 ijerph-22-00904-t002:** Model coefficients—weight regain.

Predictor	Estimates	Standard Error	Z	*p*
Intercept	21.8358	1533.045	0.01424	0.989
Negative Affect				
Light–Serious	−0.9286	2.09616	−0.44298	0.028
Moderate–Severe	−1.4397	2.08436	−0.69069	0.04
No Change–Serious	−0.5434	2.22312	−0.24443	0.807
Detachment				
Light–Serious	−17.0246	1533.037	−0.01111	0.011
Moderate–Severe	−19.5156	1533.038	−0.01273	0.99
No Change–Serious	−16.2664	1533.037	−0.01061	0.992
Antagonism				
Light–Serious	−1.2232	2.52027	−0.48534	0.027
Moderate–Severe	−3.1451	3.88871	−0.80878	0.419
No Change–Serious	−2.1592	2.56315	−0.84241	0.4
Disinhibition				
Light–Serious	−2.5410	1.04622	−2.42871	0.015
Moderate–Severe	−1.2197	0.74655	−1.63381	0.102
No Change–Serious	−0.3032	0.8382	−0.36172	0.718
Psychoticism				
Moderate–Mild	0.5662	1.33696	0.42349	0.672
No Change–Light	0.0475	0.73897	0.06433	0.949
BITE Classification				
No Risk–Risk	0.2965	0.86806	0.34151	0.733
ECAPc				
Moderate CAP–Severe CAP	−2.6886	0.74776	−3.59558	<0.001
No CAP–Severe CAP	−1.0371	0.90375	−1.14760	0.251
DASS—Depression				
Moderate–Mild	1.128	1.18343	0.95318	0.041
Minimum–Light	0.2948	1.26682	0.23272	0.816
DASS—Anxiety				
Light–Serious	0.9172	2.20877	0.41524	0.678
Moderate–Severe	1.7079	2.17996	0.78344	0.433
Minimum–Serious	3.0541	2.26328	1.34943	0.177
DASS—Stress				
Moderate–Mild	−13.7498	2399.546	−0.00573	0.995
Minimum–Light	1.59	1.51714	1.04802	0.295
Max Preoperative Weight	0.037	0.02213	1.67162	0.095
Minimum Postoperative Weight	−0.0903	0.03874	−2.33082	0.02
Age	−0.0680	0.03191	−2.13111	0.033
Cb Time (Months)	0.0258	0.0092	2.80618	0.005

Note: Estimates represent the log odds of “Weight Regain = Yes” vs. “Weight Regain = No”. Abbreviations: ECAPc refers to the clinical classification of the Periodic Binge Eating Scale; BITE stands for the Bulimic Investigatory Test of Edinburgh; DASS refers to the Depression, Anxiety, and Stress Scale—Short Form; CB Time indicates the time elapsed since bariatric surgery (in months); and CAP refers to pathological eating behavior.

**Table 3 ijerph-22-00904-t003:** Poisson regression for predictors of weight regain after bariatric surgery.

Variable	Prevalence Ratio (PR)	95% CI	*p*-Value
Binge Eating Severity	2.41	1.75–3.26	<0.001
Stress Levels	1.92	1.45–2.79	0.002
Negative Affectivity	1.67	1.29–2.24	0.004
Physical Inactivity	1.58	1.21–2.41	0.005
Anxiety	1.19	0.87–1.63	0.214
Psychoticism	1.12	0.83–1.57	0.278

## Data Availability

The original contributions presented in this study are included in the article. Further inquiries can be directed to the corresponding author.

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
