# Peer review of "Statistical and Machine Learning Modeling of Psychological, Sociodemographic, and Physical Activity Factors Associated with Weight Regain After Bariatric Surgery"

_ijerph, 2025, doi:10.3390/ijerph22060904_

Round 1
Reviewer 1 Report
Comments and Suggestions for Authors
Comments Attached, Please

Author Response
We would like to express our sincere gratitude to the reviewers and editors for their valuable feedback and constructive suggestions. All comments were carefully considered and addressed in detail. The manuscript has been revised accordingly, and a point-by-point response to each observation has been provided in the attached document.
Please see the attachment.

Reviewer 2 Report
Comments and Suggestions for Authors
Please include the validation process for the used study questionnaire.
Clearly mention the inclusion/exclusion criteria for the enrolled study participants.
The ethical approval number/code for the study is needed to be mentioned.
Please include the following three references in the discussion section:
Noria SF, Shelby RD, Atkins KD, Nguyen NT, Gadde KM. Weight Regain After Bariatric Surgery: Scope of the Problem, Causes, Prevention, and Treatment. Curr Diab Rep. 2023 Mar;23(3):31-42. doi: 10.1007/s11892-023-01498-z. Epub 2023 Feb 8. PMID: 36752995; PMCID: PMC9906605.
Athanasiadis DI, Martin A, Kapsampelis P, Monfared S, Stefanidis D. Factors associated with weight regain post-bariatric surgery: a systematic review. Surg Endosc. 2021 Aug;35(8):4069-4084. doi: 10.1007/s00464-021-08329-w. Epub 2021 Mar 1. PMID: 33650001.
King WC, Hinerman AS, Belle SH, Wahed AS, Courcoulas AP. Comparison of the Performance of Common Measures of Weight Regain After Bariatric Surgery for Association With Clinical Outcomes. JAMA. 2018 Oct 16;320(15):1560-1569. doi: 10.1001/jama.2018.14433. PMID: 30326125; PMCID: PMC6233795.
Author Response

(The authors gave the same response as above.)

Reviewer 3 Report
Comments and Suggestions for Authors
This is a well-written cross-sectional study examining the factors associated with recurrent weight gain following metabolic bariatric surgery. The study stands out for its methodological rigor, for which the authors should be commended.
One point I would like to raise concerns the selection of the factors investigated. Were these factors identified through a classification algorithm, or were they selected arbitrarily by the authors? For example, why were potential contributors, such as technical failure of the surgical procedure or patient non-adherence to follow-up visits, not included?
You can find the remainder of my comments in the attached PDF file. Most of them pertain to terminology—specifically, aligning it with current trends in our understanding of the pathophysiology of obesity—and citation updates.
I look forward to your response.

Author Response

(The authors gave the same response as above.)

Reviewer 4 Report
Comments and Suggestions for Authors
This manuscript addresses a highly relevant and timely issue: the identification of factors associated with weight regain after bariatric surgery. Given the high percentage of patients who experience weight regain post-surgery, understanding the variables involved is essential for developing preventive strategies. In this regard, the development of a predictive formula, as proposed by the authors, adds considerable value to the field and has important clinical implications.
The article is generally well-written and structured. However, I would like to offer some suggestions for improvement to further enhance the quality and clarity of the manuscript:
Abstract and Keywords
-
The abstract could be strengthened by concluding with some practical implications of the findings.
-
It is suggested that some of the more relevant psychological variables identified in the study (e.g., binge eating, stress, physical inactivity) be included as keywords instead of general terms or prevalence ratio.
Introduction
-
The concept of emotional eating, which is a key factor in weight regain after bariatric surgery, should be clearly defined and discussed.
-
Some statements in the Introduction would benefit from appropriate references to support them. For example, the two sentences in lines 68–72 lack citations. Similarly, the claims made in lines 77–78 and 83–85 are not supported by references, which undermines their scientific credibility.
-
The classification of "psychological, sociodemographic and physical activity factors" may need revision. Since physical activity is a behavior, grouping it under "behavioral factors" might be more appropriate.
Materials and Methods
-
The Methods section would benefit from the inclusion of subheadings such as Participants, Instruments, Procedure, Study Design, and Data Analysis for improved organization and readability.
-
A brief explanation of the STROBE and CHERRIES protocols would help readers unfamiliar with these reporting standards.
-
The study was conducted in 2020. The authors could explain the delay in publication.
-
Instruments should be grouped and described according to the type of variables assessed (e.g., demographic, psychological, biological) and presented in a logical order.
-
The rationale for using a bulimia-specific instrument should be clarified. It is currently unclear why the BITE was chosen.
-
It is not entirely clear what the BITE and ECAP instruments specifically assess. A detailed explanation of these instruments, including their subscales, is necessary; otherwise, the interpretation of the results becomes difficult. The same applies to the other instruments used. For instance, it is unclear how the constructs "antagonism," "negative affectivity," and "detachment" are defined within the context of the tools applied in this study. These psychological dimensions can vary significantly in meaning depending on the instrument, and therefore their definitions should be clearly stated to avoid misinterpretation.
-
To assess whether the statistical tests used are appropriate, it is necessary to know whether the variables assessed are dichotomous or continuous.
-
When referring to "socioeconomic variables," the specific variables measured should be clearly listed.
-
In line 139, the authors might have intended to say "data on height and weight" instead of "height and body mass," as height and weight are needed to calculate BMI.
Results
-
A more detailed explanation of the findings in Table 1 would help the reader understand what is reported in lines 190–191.
-
As previously noted, the exact meanings of terms like “antagonism” and “disinhibition” must be clarified to interpret the tables and figures.
-
Some of the content currently included in the Results section would be more appropriately placed in the Discussion, as it goes beyond the objective presentation of findings and begins to interpret or explain the results. For example, the information in lines 235–239 and 263–264 includes interpretive commentary that should be reserved for the Discussion section, where it can be contextualized and supported by relevant literature.
-
Figure 1 is difficult to interpret without a clear understanding of what the ECAP instrument measures. Specifically, it is unclear what PAC refers to, and how the construct of pathological eating behavior is defined in the context of this study. Does this construct include binge eating? It seems inconsistent that binge eating—typically considered a form of pathological eating behavior—was not associated with weight regain, while PAC was. This discrepancy should be clarified.
-
In the formula used to calculate the risk of weight regain, the meanings of the letters L and H are not explained. Additionally, the text refers to variables labeled C and A, but these do not appear in the actual formula. This inconsistency should be addressed and clarified.
Discussion
-
There appears to be a contradiction: while binge eating is referred to as "the strongest factor identified," the logistic regression did not find it to be significantly associated with weight regain. This discrepancy should be addressed and explained.
-
Similarly, the statement that "negative emotional stages, such as anxiety and depression" are important is contradicted by the finding that anxiety and stress symptoms were not statistically associated with weight regain. This needs clarification.
-
In the Discussion, beyond summarizing the main findings, the authors should relate their results to the existing literature. The associations between weight regain and various factors—such as binge eating and stress—are scarcely discussed in relation to previous studies, which limits the depth and contextualization of the findings.
-
The lack of association between anxiety and weight regain should be addressed, particularly as this finding appears to contradict previous research on the relationship between anxiety, emotional eating, and weight regain.
-
Some parts of the Discussion are redundant and should be streamlined, especially between lines 394–405 and 469–482.
Formal and Formatting Issues
-
Statistical terms should follow standard conventions. For example, sample size should be written in lowercase italics (n), and p values should be italicized.
-
References do not seem to conform to the journal’s required format. The authors should ensure full compliance. For example, IJERPH requires the following style for journal articles:
-
Author 1, A.B.; Author 2, C.D. Title of the article. Abbreviated Journal Name Year, Volume, page range.
-
-
Article titles should not be italicized; “vol,” “no,” and “pp.” should not be used; and the publication year should not appear at the end.
-
Table titles should be revised to better reflect their content. For instance, Table 1 presents group comparisons between participants with and without weight regain, not participant characteristics.
-
There should be a clear break between one Table and the text that introduces the following Table to avoid the appearance of a footnote.
-
The terminology in Table 2 should be consistent with the text (e.g., "light" vs. "mild").
-
Acronyms in Table 2 should be explained in a footnote.
-
Figure 1 should include a note explaining the meaning of ECAPc. It is unclear if CAP and PAC are the same, and if so, the acronym should be consistent throughout the manuscript.
-
In Figure 5, the title appears twice. The duplicate should be removed.
To sum up, this is a valuable and relevant study with significant potential impact. With improvements in clarity, structure, and adherence to academic standards, the manuscript can make a strong contribution to the literature on weight regain after bariatric surgery. I encourage the authors to consider the above suggestions in their revisions.
Author Response

(The authors gave the same response as above.)

Round 2
Reviewer 1 Report
Comments and Suggestions for Authors
No More Comments